# Microbial induced calcite precipitation can consolidate martian and lunar regolith simulants

**Rashmi Dikshit**[1], **Nitin Gupta**[1], **Arjun Dey**[2], **Koushik Viswanathan**[1], **Aloke Kumar**[1]*

**1** Department of Mechanical Engineering, Indian Institute of Science, Bangalore, India, **2** Thermal Systems Group, U. R. Rao Satellite Centre (Formerly ISRO Satellite Centre), Bangalore, India

* alokekumar@iisc.ac.in

**Data Availability Statement:** All relevant data are within the paper and Supporting Information files.

**Funding:** RD BT/PR31844/BIC/101/1206/2019 Department of Biotechnology, Ministry of Science and Technology, GOI https://dbtindia.gov.in/ The

## Abstract

We demonstrate that Microbial Induced Calcite Precipitation (MICP) can be utilized for creation of consolidates of Martian Simulant Soil (MSS) and Lunar Simulant Soil (LSS) in the form of a 'brick'. A urease producer bacterium, *Sporosarcina pasteurii*, was used to induce the MICP process for the both simulant soils. An admixture of guar gum as an organic polymer and $NiCl_2$, as bio- catalyst to enhance urease activity, was introduced to increase the compressive strength of the biologically grown bricks. A casting method was utilized for a slurry consisting of the appropriate simulant soil and microbe; the slurry over a few days consolidated in the form of a 'brick' of the desired shape. In case of MSS, maximum strength of 3.3 MPa was obtained with 10mM $NiCl_2$ and 1% guar gum supplementation whereas in case of LSS maximum strength of 5.65 Mpa was obtained with 1% guar gum supplementation and 10mM $NiCl_2$. MICP mediated consolidation of the simulant soil was confirmed with field emission scanning electron microscopy (FESEM), X-ray diffraction (XRD) and thermogravimetry (TG). Our work demonstrates a biological approach with an explicit casting method towards manufacturing of consolidated structures using extra-terrestrial regolith simulant; this is a promising route for *in situ* development of structural elements on the extra-terrestrial habitats.

## 1. Introduction

Human curiosity to explore the deep mysteries of space may end up eventually demanding the development of extra-terrestrial settlements. Mars and moon, being close to Earth, have been identified as unprecedented sources of processable materials and have thus become the preferred choices for space organizations around the world to build temporary structures and observatories [1]. On the occasion of the 50th anniversary of manned mission landing on lunar surface, space agencies like the National Aeronautics and Space Administration (NASA) and the European Space Agency (ESA) announced plans to restart manned missions for the exploration of outer space [2–6]. For feasible and sustainable space exploration, these habitats need to be built from available *in situ* resources on moon/Mars [7–11]. Within space lexicon, the

funders had no role in study design, data collection and analysis, decision to publish, or preparation of the manuscript. AK ISTC0415 Indian Space Research Organisation https://www.isro.gov.in/ The funders had no role in study design, data collection and analysis, decision to publish, or preparation of the manuscript.

**Competing interests:** The authors have declared that no competing interests exist.

term *in situ* resource utilization (ISRU) refers to any process that encourages processing of local resources found during exploration of extra-terrestrial habitats in order to reduce dependency on materials chaperoned from earth. Both martian and lunar surfaces have an abundance of fine soil on their surfaces, termed regolith, which can be utilized as building or raw construction material [1, 3]. The primary challenge in this endeavour lies in the consolidation of the unbonded fine regolith particles into a structure of substantial mechanical integrity.

Towards this end, various methodologies for consolidating regolith have been proposed. Fusing of regolith particles through laser-sintering [11], use of microwave irradiation for bonding [12, 13], production of cast basalt [9, 14–16], fabrication with lunar glass [17, 18], creation of sulfur-based concrete [9, 19], dry-mix/steam injection methods and various 3D printing methodologies [20–22] are examples of processes that have been proposed. Some of these approaches in more detail are discussed here. In one work, structure produced by lunar regolith prepared by mixing of lunar regolith simulant (JSC-1, JSC-1AF, and JSC-1AC) and sintered at 1200˚C for 20 min delivered compressive strength of the order of 103.2 to 232 MPa [23]. In another work, researchers have proposed the possibility of making structural units directly with the unrefined sintered lunar regolith. Two batches of this regolith simulant JSC-1A with different porosities were used yielding average compressive strength of 218.8 MPa, and 84.6 MPa respectively [24]. Despite this significant strength, the sintering process is plagued by several inherent drawbacks such as high porosity, thermal cracking and difficulty in casting larger samples [24]. Also, clustering and coalescence of pores while sintering in microgravity makes the material less strong as compared to that sintered on earth [3, 25]. Work carried out by Young-Jae Kim et. Al [26], demonstrated consolidation of KLS-1 lunar regolith simulant with microwave sintering method at 1120˚C and compressive strength reported was approximately 39 MPa. Photocurable lunar regolith simulant suspension was developed to create small regolith bound structures with utilization of vat polymerization (VP) 3 D method. These 3D printed structures achieved the compressive strength of approximate 5 MPa [27]. In a recent development, biological approach where human serum albumin (HSA), a protein extracted from human blood plasma have been explored to make a structural unit of lunar and martian regolith. HSA was used as binder to fabricate the consolidated structure and gave compressive strength of the order of 20 to 32 MPa [28].

Given the importance and burgeoning interest in extra-terrestrial human settlements, it is necessary that various other methods of creating structures from lunar/martian regolith be probed which are feasible, economical and practicable in nature. With this intent, we discuss here a casting method for fabrication of brick-like structures using regolith simulants and a biomineralization process called microbial induced calcite precipitation (MICP).

MICP is a biomineralization process that produces calcium carbonate by exploiting the metabolic activity of bacteria [29–33] via various pathways [34]. The urease pathway is widely explored [35–38] wherein the conditions for mineral precipitation are made favourable by controlled reactions involving hydrolysis of urea by ureolytic bacteria. Urease (E.C. 3.5.1.5), which is a nickel dependent and non-redox enzyme is primarily responsible for urea hydrolysis [39, 40]. One possible route for enhanced activation of urease activity is by changing the concentration of Ni (II) ions. Crystal structures of microbial urease enzyme studied from *Klebsiella aerogenes* [41] and *Bacillus pasteurii* [42] organisms suggest that it has divalent nickel ion at the centre, which aids in binding the substrate (urea), alleviates the catalytic transition state thus accelerating the ureolysis reaction rate [43].

In this work we explore this property to enhance the use of MICP based consolidation with martian and lunar regolith. Our study shows a significant increase in the compressive strength of the resulting consolidated brick-like structures with both martian and lunar regolith simulants. The present work also utilizes a naturally occurring and economically viable bio-polymer

guar gum, to further improve the strength of these bio-consolidated 'space bricks.' Guar gum acts as a binder [30, 44] with soil for improving mechanical strength, and is also stable with pH/temperature variation [45, 46] thus making it an ideal additive for MICP mediated space brick formation. For our work, we used the microbe *Sporosarcina pasteurii*—a much explored MICP capable bacteria and a gram positive non-pathogenic strain [33, 47–49]. We have also designed a modular self-contained lab-on-a-chip (LoC) device for the real time monitoring of the MICP process and demonstrated the survival of this organism under miniaturized experimental environment [50]. We believe that such studies will help evaluate the possibilities of MICP as a sustained solution for building extra-terrestrial settlements utilizing in-situ resources.

## 2. Materials and methods

### 2.1 Microorganisms and culture conditions

Bacterial-induced bio-consolidation was explored using ureolytic bacterial strain; namely, *Sporosarcina pasteurii* (Miquel) Yoon *et al.* [51] ATCC®11859™, procured from American Type Culture Collection (ATCC) and revived using ATCC recommended media ($NH_4$-YE liquid medium).

Flask condition experiments were performed in four different sets as follows:

1. Synthetic media (hereafter, SM); prepared with 0.1 g glucose, 0.1 g peptone, 0.5 g NaCl, 0.2 g mono-potassium phosphate and 3 g urea in 100 ml of distilled water.

2. Synthetic media-guar gum (hereafter, SM-GG); prepared by replacing glucose in SM medium with 1% (w/v) guar gum (Urban Platter, India).

3. Synthetic media-$NiCl_2$ (hereafter, SM-N); prepared by adding 10 mM $NiCl_2$ in SM medium.

4. Synthetic media-GG-$NiCl_2$ (hereafter, SM-GG-N); prepared by adding 10 mM $NiCl_2$ in SM-GG medium.

Urea was filter sterilized (0.25μm) and added to the medium after autoclaving at 121˚C and 15 psi for 30 minutes to prevent degradation. Medium was inoculated with 5% culture of *S. pasteurii* grown up to log phase with 0.8 OD and further incubated at 30˚C. Parameters such as optical density (using UV/Vis spectrophotometer, Shimadzu, Japan) at 620 nm wavelength, pH (CyberScan pH meter, Eutech Instruments) and quantification of ammonium ion concentration using Nessler's reagent assay [37] were recorded at different time intervals. The data was recorded in triplicates and mean value was plotted. All chemicals were procured from Hi-Media, India and used without further purification.

Gas Chromatography Mass Spectrometry (GCMS) profiling was performed for metabolites mapping produced with and without guar gum supplementation in the SM medium. For GCMS (GC column: Agilent 7890A MS: 5975C MSD Electron Impact Ionization Mass Analyze equipped with Quadrupole Software), 1 μL of methanol extract of bacterial metabolites was injected into the GC inlet. Mass spectral library (NIST2011) was used to identify the fraction based on their mass spectra (m/z ratio).

### 2.2 Characterization of unconsolidated (raw) soil simulants

Martian soil simulant (MSS) procured from Class Exolith lab Florida [52], and lunar soil simulant (LSS) developed by the Indian Space Research Organisation (ISRO) [53] were used for the fabrication of the martian and lunar bricks. Particle size distributions for both soil simulants

were measured by suspending 0.1g soil in 5ml ultrapure distilled water, followed by sonication to disperse the particles. After sonication, particle suspension was placed on a glass bottom petri dish (ibid) and imaged using Leica DMI 8 optical microscope (Germany). The images were captured using a Leica DMi8 inverted microscope with Leica DFC3000G camera and a 10x objective lens. The resulting image resolution was 1 micrometer per pixel. Particle size histograms were obtained using a 50 micrometer bin size. The images were processed using a standard MATLAB® (Mathworks) routine (regionprops) to determine particle dimensions by fitting a horizontally aligned rectangle to each particle. The minimum equivalent diameter for LSS was 7.44 micrometers and for MSS particles was 6.62 micrometers. In order to avoid erroneous counting of small insignificant trace particles, the minimum major length of the fit rectangle was selected as 25 pixels (25 μm). Particle sizes were obtained from the corresponding rectangles as the radii of equivalent circles with the same area. Correspondingly, the minimum equivalent diameter for LSS was 7.44 μm and for MSS particles was 6.62 μm.

## 2.3 Casting process for repeatable and scalable sample preparation

In lieu of a conventional bioreactor, for the present work we designed and used aluminium molds to cast bricks of uniform size and shape, in a repeatable and scalable manner. The molds were made of aluminium alloy (Al6061-T6), machined using a vertical milling machine. Each mold was made in two mating parts, to enable easy casting and parting off, and consisted of five cuboidal cavities with a cross section of 32 x 32 mm$^2$ and a height of 35 mm. Thus, a single two-part mold could be used to cast five samples simultaneously. A schematic representation of this casting process for MICP mediated consolidation is depicted in Fig 1 and representative image of space bricks are given in the Fig 2. The inner surface of the mold was covered with a thin transparent plastic (OHP) sheet to aid easy removal of consolidated samples from the mold. Fifty grams of autoclaved simulant soil (at 121˚C and 15 psi for 30 minutes) were mixed with media and various combinations of treatments as presented in Table 1, and discussed in Sec. 2.1. S. pasteurii with optical density at 620 nm ($OD_{620}$) 1.5 in $NH_4$-YE medium was used as inoculum in all the treatments. The soil-bacteria-medium mixture was tightly packed in the mold cavities and the upper part of the mold was sealed with parafilm strip, see Fig 1. Incubation period was set for 5 days at 32˚C followed by drying in a hot air oven (BioBee, India) at 50˚C for a period of 24 h.

## 2.3 Mechanical and materials characterizations

Compression testing of consolidated MSS and LSS samples was carried out on a Universal Testing Machine (Instron-5697), with a 5 kN capacity load cell and loading rate of 1 mm/min. To ensure uniform compression during the test, all surfaces of the cubical samples were grinded and polished using a portable grinder (BOSCH GWS 600). The final specimen for testing was in the form of a cube with dimensions 25±2 mm. A minimum of three test specimens were made for each set of treatments, and the average value of the data was plotted. All samples were tested in same condition except for MSS-SP-N and MSS-SP-GG-N. In the latter two cases, 2 mm rubber sheets were used between the sample surface and the UTM grips to uniformly distribute the load across the sample surface. This was because making perfectly flat samples for these two conditions proved to be extremely challenging, even after multiple grinding and polishing attempts. Furthermore, given the nonuniform nature of consolidated soil samples in general, the measured stress-strain curves contained a few small intermediate peaks. Peaks with a maximum peak/valley distance of less than 6% of the overall load maximum were considered in the results reported and more than 6% of overall load maximum were considered as the failure of the samples.

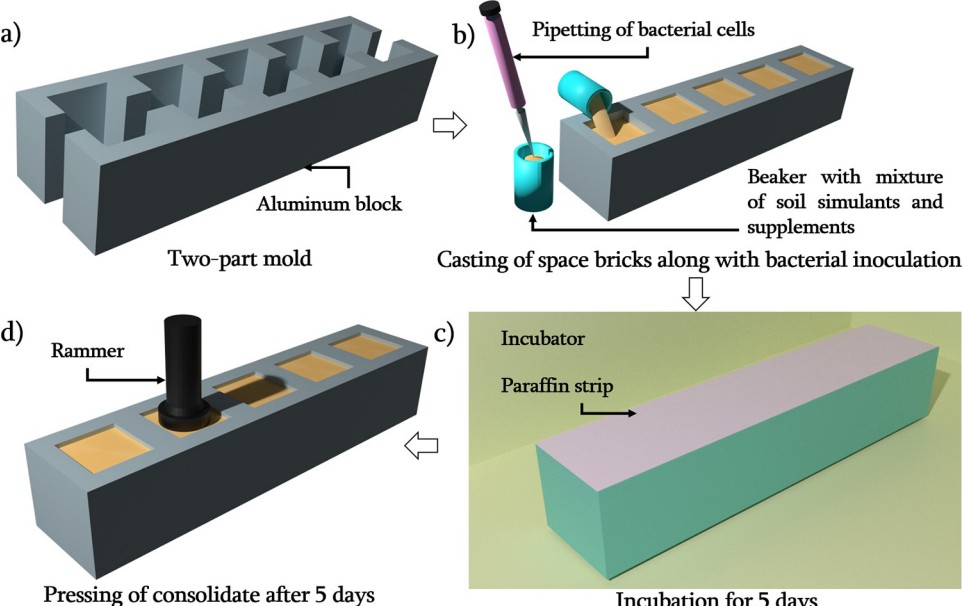

**Fig 1. Schematic representation of casting of aluminium mold and process involved in MICP mediated consolidation of simulant soils.**

The microstructure of bio-consolidated martian and lunar bricks was observed using field emission scanning electron microscopy (FESEM: Carl Zeiss AG—ULTRA 55, Germany). Different calcium carbonate phases were identified using X-ray diffractometer (XRD: PANalytical Philips diffractometer) under Cu-Kα (λ = 1.54 Å) X-ray radiation and thoroughly indexed as

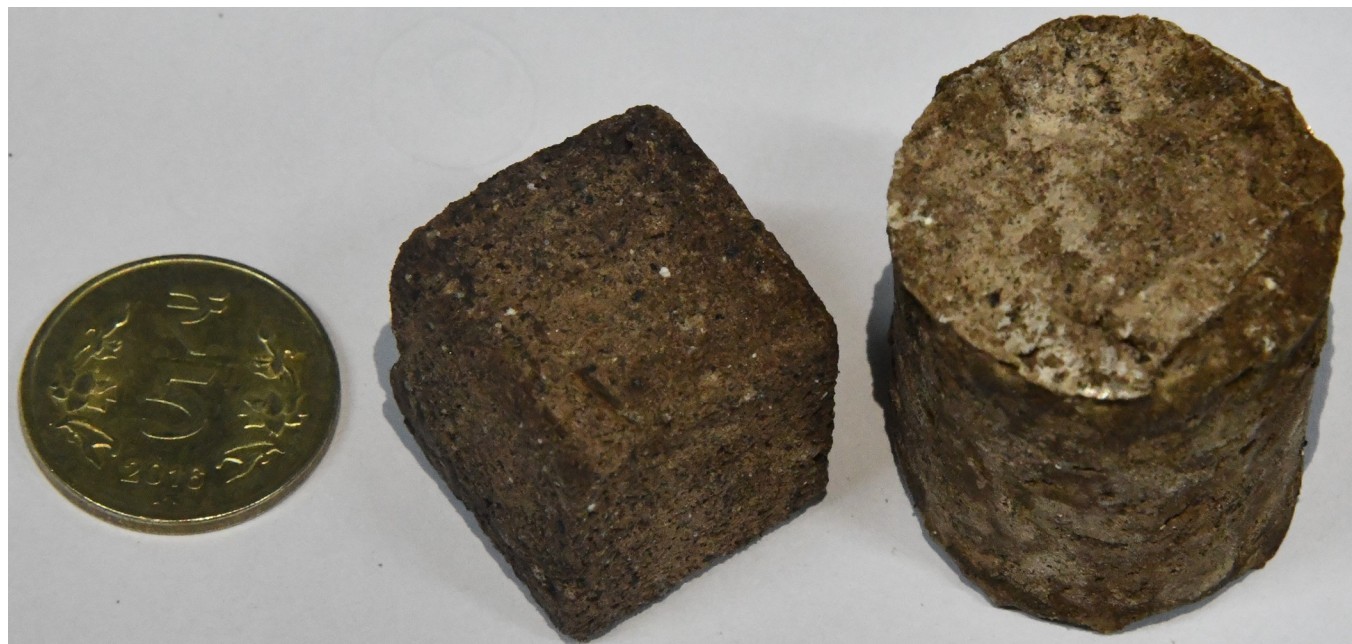

**Fig 2. Representative image of consolidated samples using MSS.** Such structures have been termed space bricks.

**Table 1. Treatments used for bacterial induced consolidation of soil simulants.**

| Name | Precipitation Media | Calcium Lactate | Guar Gum (% w/w) | NiCl$_2$ (mM) | Bacterial Strain |
|---|---|---|---|---|---|
| MSS-SP | SM-U | 50 mM | 0 | 0 | *Sporosarcina pasteurii* |
| MSS-SP-GG | SM-U | 50 mM | 1 | 0 | *Sporosarcina pasteurii* |
| MSS-SP-N | SM-U | 50 mM | 0 | 10 | *Sporosarcina pasteurii* |
| MSS-SP-GG-N | SM-U | 50 mM | 1 | 10 | *Sporosarcina pasteurii* |
| LSS-SP | SM-U | 50 mM | 0 | 0 | *Sporosarcina pasteurii* |
| LSS-SP-GG | SM-U | 50 mM | 1 | 0 | *Sporosarcina pasteurii* |
| LSS-SP-N | SM-U | 50 mM | 0 | 10 | *Sporosarcina pasteurii* |
| LSS-SP-GG-N | SM-U | 50 mM | 1 | 10 | *Sporosarcina pasteurii* |

per Inorganic Crystal Structure Database (ICSD) library using PANalytical X'Pert High Score Plus pattern analysis software.

For thermogravimetry (TG) (Mettler Toledo TGA analyser STARe SW10.00,) analysis of space bricks approximately 10 mg soil samples were taken from each treatment separately and placed on a cylindrical 70 μL alumina crucible followed by heating from 0˚C to 1000˚C. The heating rate was set at 10˚C/min throughout the experiment.

## 3 Results and discussions

Lunar and martian soil/regolith simulants used in this experiment were drawn from two different sources and their mineral composition/particle size distribution was first established.

### 3.1 Lunar and martian soil simulants: Composition and particle properties

Fig 3 depicts the size and morphological characterization for MSS and LSS particles. Fig 3A and 3B show the size histogram for MSS and LSS particles, respectively, with insets showing corresponding representative FESEM images. The mean particle diameter for LSS particles was approximately 34 μm with a standard deviation of 26 μm, while corresponding numbers for MSS were 31 μm and 33 μm, respectively. This indicates that that MSS typically contains a wider distribution of particle sizes than LSS. However, in both cases, particles were observed to be irregularly shaped. The particle aspect ratio, determined as discussed in Sec. 2.3, quantifies this irregularity; aspect ratios for MSS and LSS particles were around 0.5 and 0.56, respectively.

In order to identify the phases and the elements present in both soil simulants, X ray diffraction (XRD) and energy dispersive spectroscopy (EDS) were performed, see Fig 4. In MSS, plagioclase, pyroxene and olivine (Fig 4 (A)) were identified as prime crystalline phases whereas in LSS, plagioclase (Fig 4 (B)) was the major crystalline phase. MSS possesses significant amount of pyroxene (XY (Si, Al)$_2$O$_6$), that contains Mg and Fe along with Si and aluminium oxide. Correspondingly, major elements observed in MSS from EDS mapping were O, Si, Al, Mg and Fe as shown in Fig 4 (C). Fe and Mg are essential metal ions required in limited concentration for bacterial growth. It is well known that Mg is important for cell division of the rod-shaped bacteria whereas iron contributes in biological activity such as electron transfer and enzymatic activity [54]. Incidentally, iron appears in two oxidation states viz. Fe$^{2+}$ and Fe$^{3+}$ in the bacterial cells that can be transformed into each other. Though these ions are essential for bacterial growth and activity, their requirement is very low. It is reported that higher concentrations of iron may lead to toxic effects on bacterial cells due to generation of reactive oxygen compounds [55]. This may cause peroxidation of lipids cell membrane, protein and can also damage cellular DNA. [56]. The relevance of these observations will be discussed

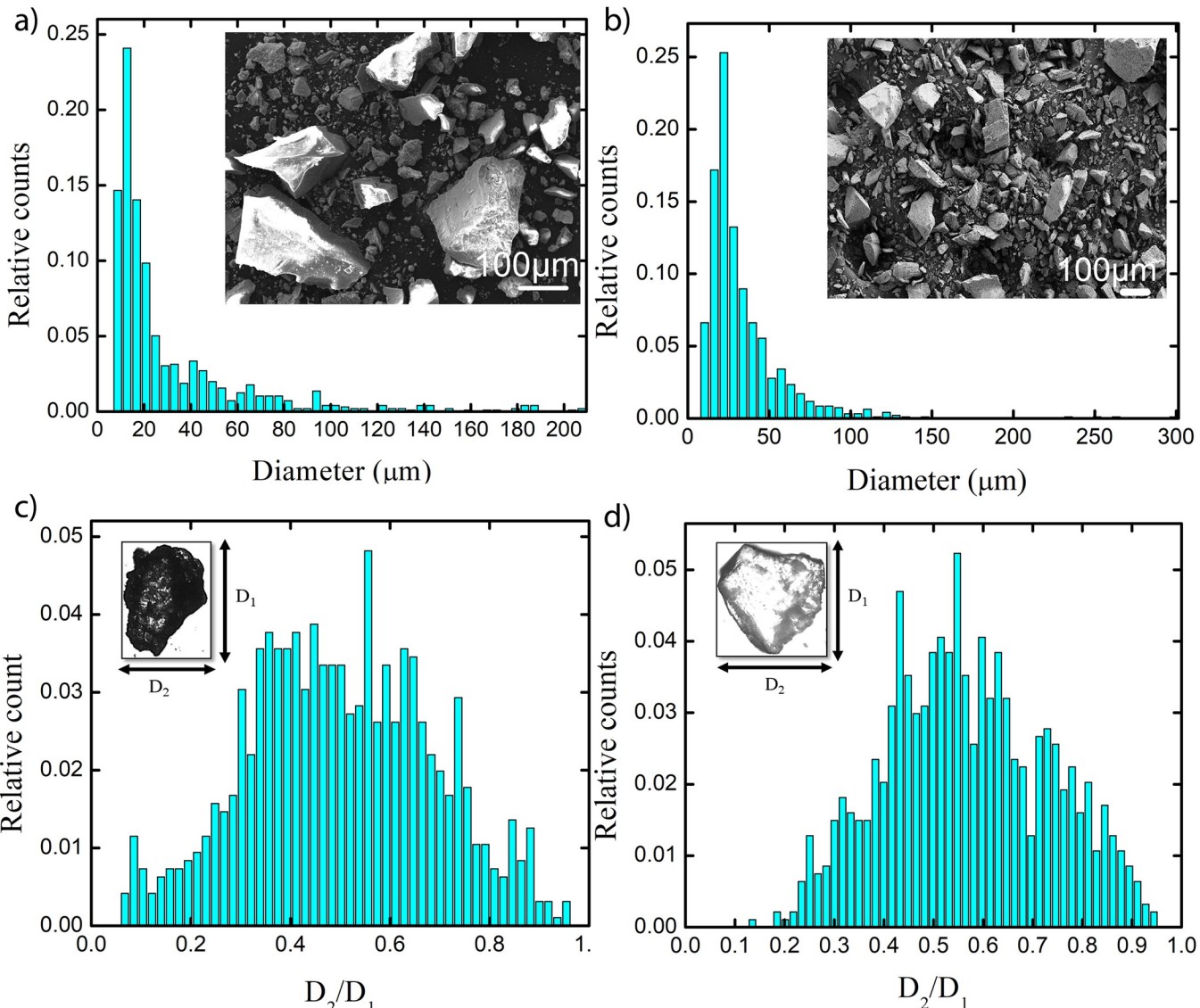

**Fig 3.** Characterization of raw MSS and LSS: Size distribution of a) MSS particles with an inset FESEM image of MSS and b). LSS particles with an inset FESEM image of LSS c) Aspect ratio for MSS particles and d) Aspect ratio for LSS particles.

subsequently when consolidation results of MSS and LSS are discussed. In the case of LSS, the major mineral identified was plagioclase, composed of $NaAlSi_3O_8$ and $CaAl_2Si_2O_8$ and containing Si, Al, Na and Ca elements. These elements were, consequently, also detected in the EDS maps presented in Fig 4(D).

### 3.2 MICP induced consolidation of martian regolith bricks

The molding/casting process adopted here allows us to make various shapes and even hollow structures that are simply not possible in a conventional bioreactor. For the purpose of determining the optimal process parameters, a mixture of microbial culture and regolith were premixed in slurry form and incubated in the mold cavities for a period of about 5 days. At the end of the process, these cubical samples were retrieved, dried and subjected to quasi-static compression tests, as described in Sec. 2.3. Fig 5A shows the results of compressive strength

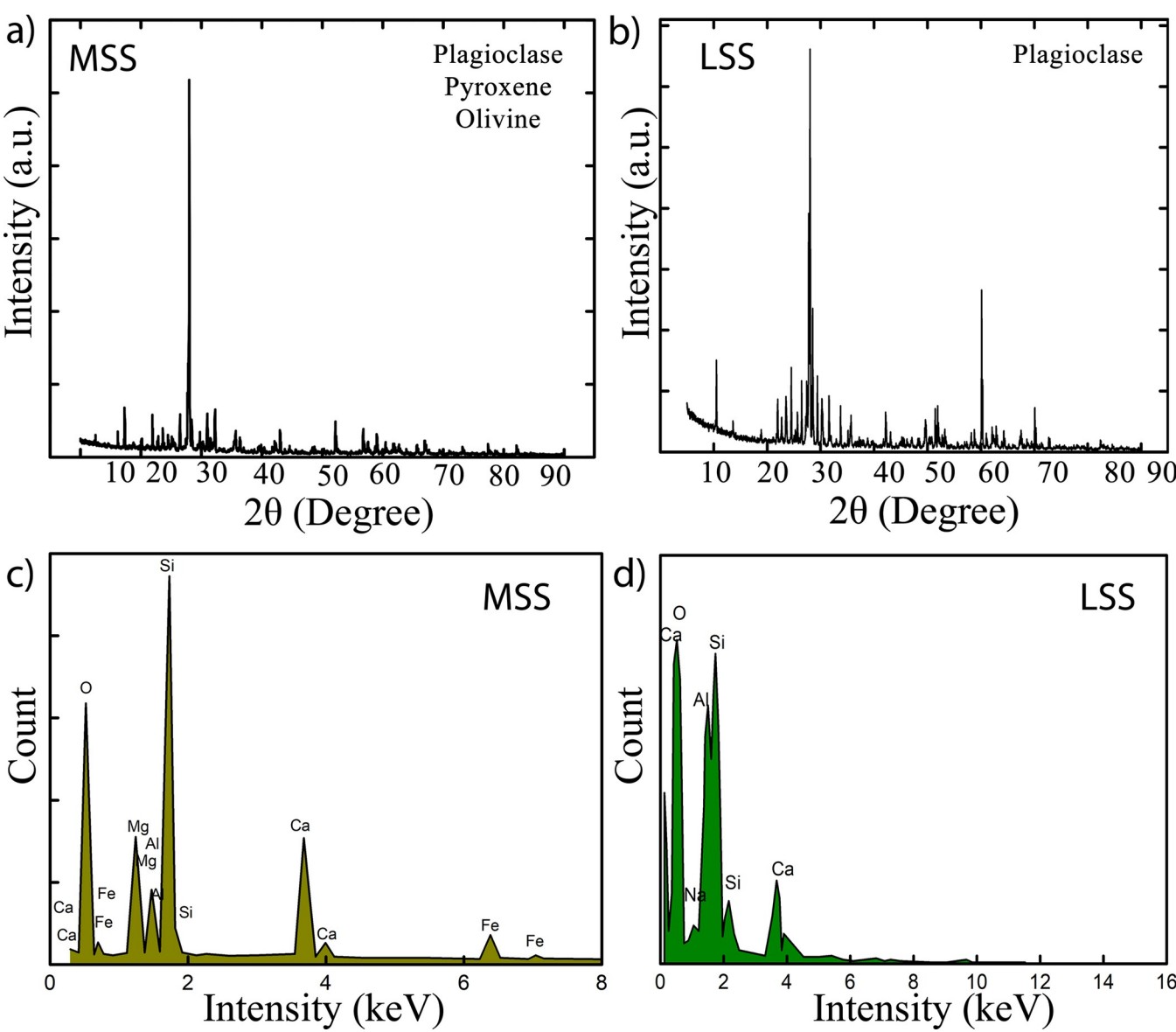

**Fig 4. XRD pattern and EDS of raw soil simulants a) identified crystalline phases of MSS b) identified crystalline phases of LSS c) elemental mapping of selected location of MSS d) elemental mapping of selected location of LSS.**

measurement on various slurries containing different media combinations. MSS slurries mixed with only *S. pasteurii* or SP culture, when retrieved and dried, were not robust enough for uniaxial testing so data corresponding to this medium (MSS-SP) is not included in Fig 5A. Non-consolidation of MSS is probably due to the presence of approximate 12% Mg and 6.9% iron causing inhibition of urease activity of *S. pasteurii* in absence of additives.

Supplementing the MSS slurry with 1% guar gum resulted in mean compressive strength of approximately 1.1 MPa, see MSS-SP-GG bar in Fig 5A.

Since the urease enzyme is a metalloenzyme containing nickel at the centre [43, 57], potential enhancement of urease activity via Ni supplementation was evaluated next. Flask growth data suggested that 10 mM of Ni supplementation can enhance urease activity significantly and thereby accelerate the overall MICP process (S1 Fig). The mean compressive strength of

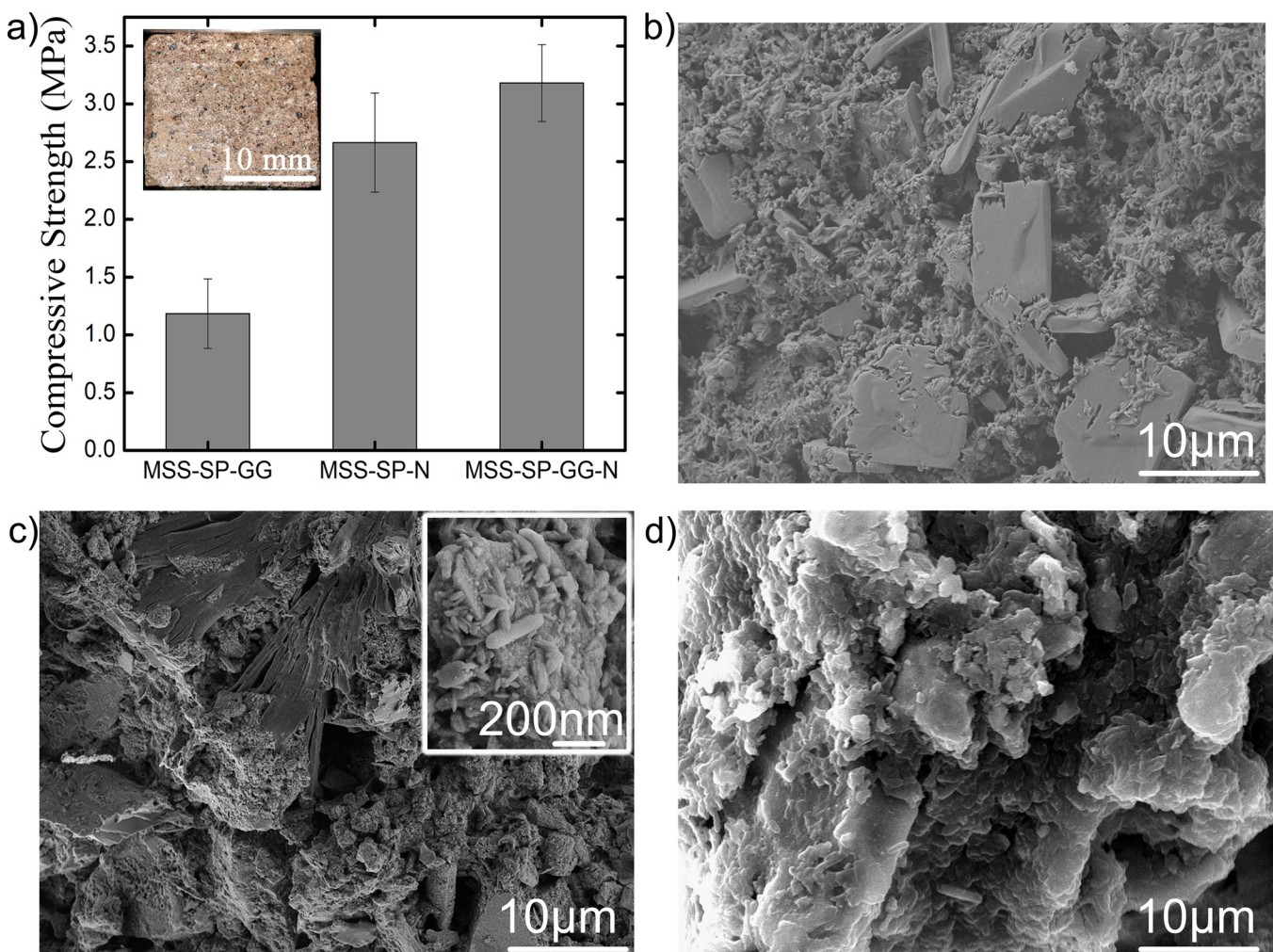

**Fig 5.** a) Compressive strengths of martian bricks for different treatments with an inset image of cubical biconsolidated martian brick. b) showing SEM micrograph of martian bricks for MSS-SP-N treatments) SEM micrograph of martian bricks for MSS-SP-GG-N treatments. d) SEM micrograph for MSS-SP-GG treatment.

the consolidated brick sample obtained by 10 mM NiCl$_2$ supplementation (MSS-SP-N) was found approximately 2.67 MPa. Hence, the strength was consistently found to be twice as large as that of the MSS-SP-GG consolidated samples. Subsequently it was deemed natural to evaluate the compressive strength of consolidated samples with both guar gum (GG) as well as Nickel supplementation. The MSS supplemented with 1% guar gum and 10 mM Nickel (MSS-SP-GG -N) showed a significantly larger mean compressive strength, of 3.3 MPa, which was the largest among all the treatments, see Fig 5(A). It is hence clear that the addition of guar gum and Ni results in significantly stronger consolidates, approaching the compressive strength of ice.

Structural changes arising in consolidated MSS due to MICP were investigated by XRD analysis see Fig 6A and supported by FESEM imaging presented in Fig 5B, 5C and 5D. The XRD patterns for the three different treatments (MSS-SP-N, MSS-SP-GG and MSS-SP-GG-N) are presented in Fig 6A. It is clear that several calcium carbonate phases (e.g., calcite, aragonite and vaterite) of precipitates are observed in all three cases. In case of MSS-SP-GG-N treatment, the peaks correspond to the calcite phase at 2θ values of 23.01˚, 29.2˚, 35.50˚, 62.7˚, 65.1˚ (hkl

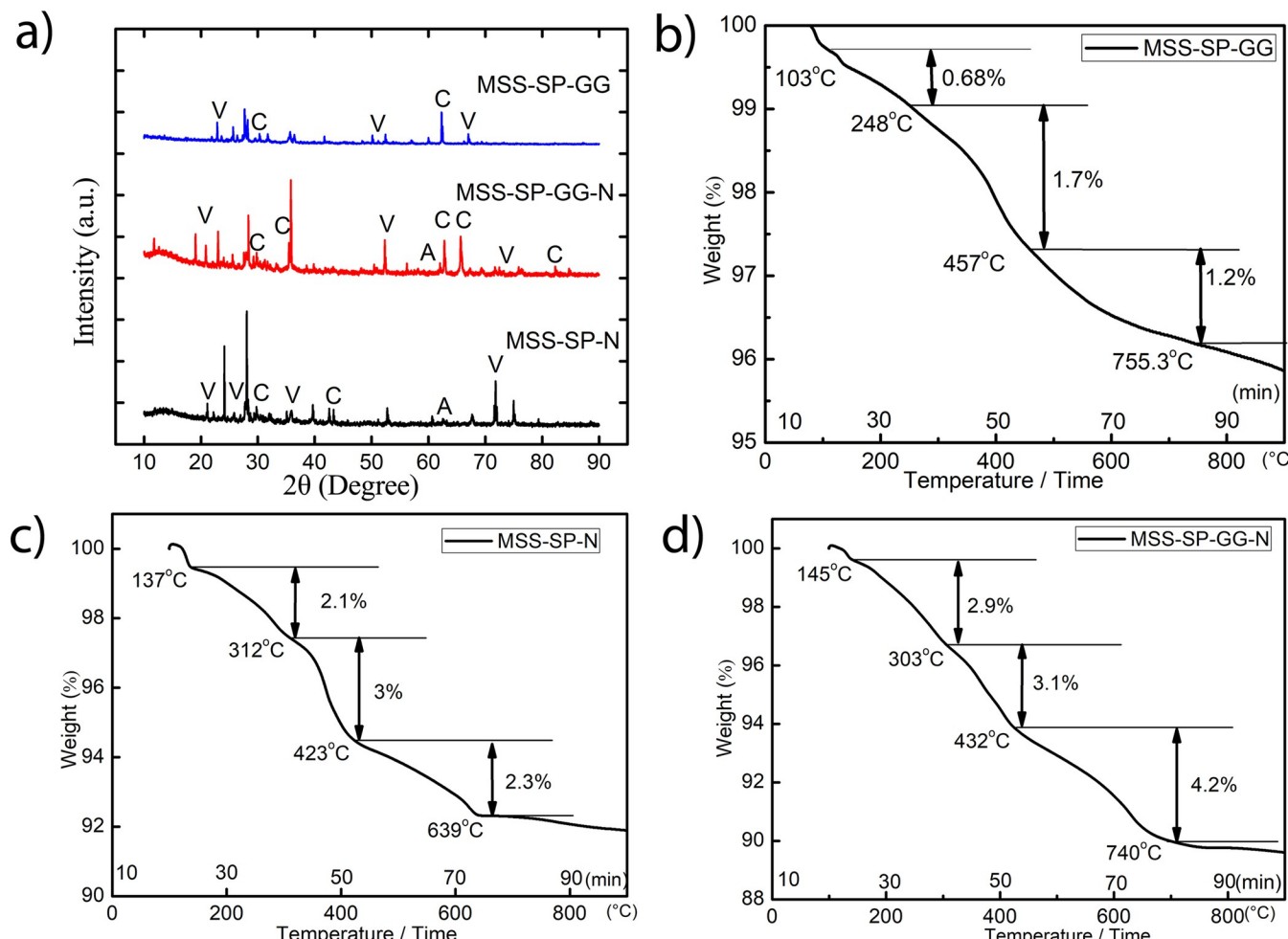

**Fig 6.** a) XRD pattern of consolidated martian bricks with different treatments. (V–vaterite; C- calcite; A- aragonite). TG curves: b) MSS supplemented with guar gum and treated with S. pasteurii c) MSS supplemented with Ni and treated with S. pasteurii d) MSS supplemented with guar gum and Ni and treated with S. pasteurii.

value 012, 104, 111, 212, 122) matched with ICSD File nos. 98-000-5337, 98-002-3930, 98-001-442 respectively whereas peaks at 2θ values of 20.8˚, 52.32˚ (ICSD File no. 98-010-9796) were observed for vaterite phase. Aragonite phase matching with ICSD File no.– 98-011-4648 was also observed at 2θ values of 67.42˚. In MSS-SP-N treatment major peaks at 2θ values 29.3˚, 42.5˚ for calcite phase with hkl (104, 002) matched with ICSD File no. 98-000-5339, 98-001-4421. Major identified peaks in MSS-SP-GG treatments were at 2θ values 62.7˚ matching with ICSD File no. 98-001-4421 for calcite phase of calcium carbonate. As expected, treatment without any additives (MSS-SP) did not show any significant calcium carbonate peaks indicating inhibition of MICP activity and, consequently, poor strength of the consolidated sample.

The FESEM micrograph of bio-consolidated MSS bricks also clearly show depicted bacterial mediated consolidation. An aggregated soil mass with appreciable bacterial induced precipitates were seen in MSS-SP-N and MSS-SP-GG-N treatment (Fig 5B and 5C) whereas in the case of MSS-SP-GG treatment bacterial induced covering on the soil mass was observed (Fig 5D).

### 3.3 MICP induced consolidation of lunar regolith bricks

An analogous procedure was followed the lunar soil simulant (LSS) and the results are summarized in Fig 7. Similar to the MSS slurry treatments, supplementation with both 1% guar gum and 10 mM $NiCl_2$ resulted in bricks with the maximum compressive strength, see Fig 7. Furthermore, guar gum and Ni supplemented LSS samples (LSS-SP-GG-N) showed mean compressive strength of 5.65 MPa, which was significantly higher than the corresponding value for MSS-SP-GG-N. Additionally, LSS consolidation using a slurry with just the microbial medium (LSS-SP) was found to be robust enough to be subjected to uniaxial testing, with a resulting mean compressive strength of 0.75 MPa. When using only guar gum supplement (LSS-SP-GG), mean compressive strength of 3.41MPa was obtained.

These results seem to suggest that MSS provides a less ideal environment for bacterial growth as compared to LSS primarily due to the existence of metals such as Fe and Mg in

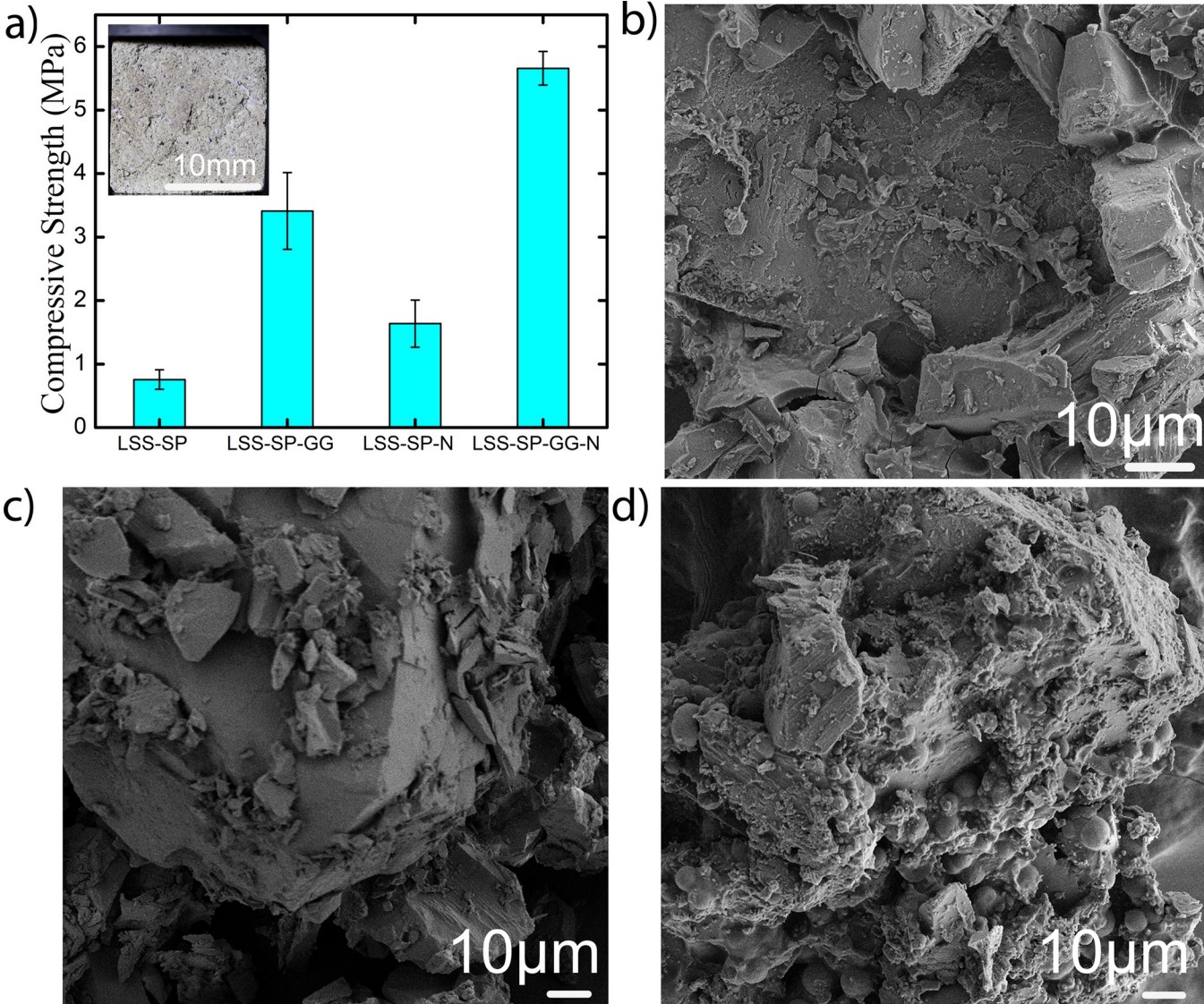

**Fig 7.** a) Compressive strengths of lunar bricks for different treatments with an inset image of cubical biconsolidated lunar brick b) showing SEM micrograph of lunar brick with nickel chloride supplementation c) SEM micrograph of lunar bricks with guar gum supplementation d) with guar gum and nickel chloride supplementation.

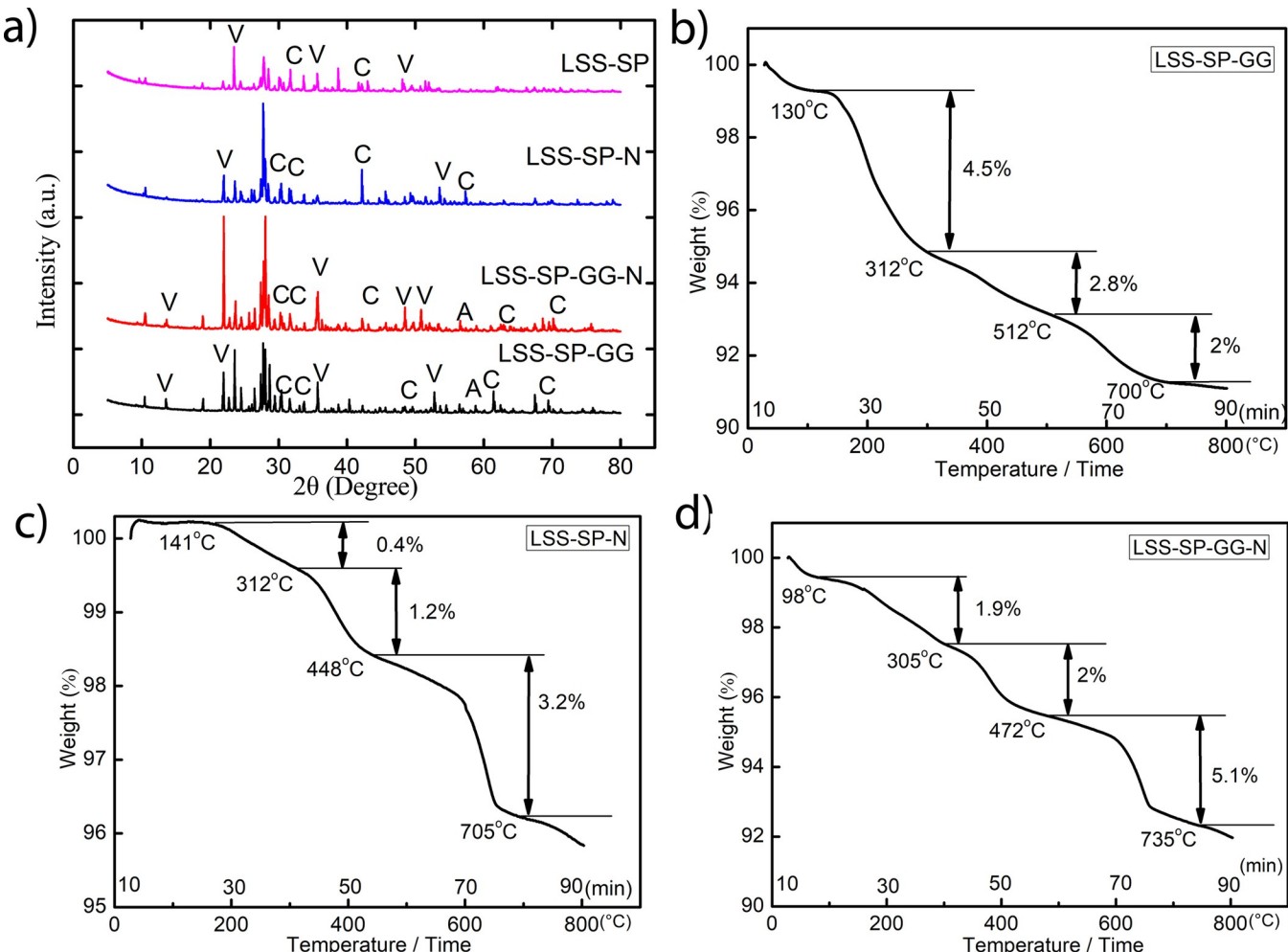

**Fig 8.** a) XRD pattern of consolidated lunar bricks with different treatments. (V–vaterite; C- calcite; A- aragonite). TG curves: b) LSS supplemented with guar gum and treated with S. pasteurii c) LSS supplemented with Ni and treated with S. pasteurii d) LSS supplemented with guar gum and Ni and treated with S. pasteurii.

significantly higher quantities. An additional difference between the consolidated LSS and MSS bricks is that LSS-SP-GG samples showed higher mean compressive strength compared to LSS-SP-N samples. The reason for this difference is at present not well understood and requires further investigation.

XRD analysis of lunar bricks showed multiple calcium carbonate peaks, as expected, thus confirming bio-consolidation (Fig 8 (A). The major identified peaks in the treatment of LSS-SP-N corresponding to the calcite phases were found at 2θ values 23.02˚ (hkl 012), 29.40˚ (hkl 104), 35.9˚ (hkl 110) and 47.0˚ (hkl 018) and matched with ICSD File nos. 00-005-0586. In the case of LSS-SP-GG-N, identified phases of calcite were at 2θ values 29.3˚ (hkl104), 35.40˚ (hkl 110), 42.01˚ (hkl 200), matched with ICSD File nos. 00-005-0586,980005339 and 98-011-4421 respectively, whereas for vaterite phase at 2θ values 45.6˚ (301), 49.6˚ (hkl 412), 53.7˚ (hkl 222) and matched with ICSD File nos. 98-010-9797.The peaks correspond to the calcite phase at 2θ values of 24.60˚ (hkl 111), 31.50˚ (hkl 202) matched with ICSD File nos. 98-007-8903, 98-009-6175 whereas peaks at 2θ values of 10.36˚ (hkl 002), 13.5˚, 65.49˚ (hkl 420) (ICSD File nos. 98-000-6092, 98-010-9797, 98-010-9796) was observed for vaterite phase.

Aragonite phase matched with ICSD File nos.– 98-011-4648, 98-011-4649 was also observed at 2θ values of 57.71˚ with hkl 122 in case of LSS-SP-GG treatment.

Just as with the consolidated MSS bricks, aggreged soil masses were also observed in the case of the LSS bricks, see FESEM micrographs in Fig 7B, 7C and 7D. In the case of guar gum supplement, dense aggregated soil mass was observed (Fig 7C) while bacterial induced matrix and precipitates covering the LSS particle were observed in the treatment with 1% guar gum and 10 mM Nickel admixture (Fig 7D).

## 3.4 Thermogravimetric (TG) characterization of space bricks

Thermal analysis for the precipitates was performed for the consolidated space bricks as shown in Fig 6B, 6C and 6D (for MSS) and Fig 8B, 8C and 8D (for LSS) consolidated bricks, respectively. The TG curves can be divided into three regions based on the transitions observed due to thermal decomposition. The first part of the curve within the range of 50 to 250˚C can be attributed to the loosely bound water or moisture content present in the samples [58]. The second part in the temperature range of 300 to 800˚C can further be divided into two stages of decarbonation. The first stage is between 300 to 500˚C where decomposition of poorly crystalline phase of $CaCO_3$ occurs and in the second stage (500 to 800˚C), decomposition of vaterite and calcite occurs [59, 60]. It can be observed from Figs 6C and 8C that the supplementation of Ni reduces the decomposition of the first poorly crystalline phase. It appears that Ni supplementation (Ni being a central part of the urease enzyme) accelerates the formation of crystalline phases of $CaCO_3$ in both the soil simulants as compared to guar gum supplementation alone. Maximum decomposition with both soil simulants was observed in the range of 500 to 700˚C (around 3.2% in case of LSS-SP-N and 3% in case of MSS-SP-N) which supports the formation of calcite and vaterite phases. However, this crystalline phase was further extended in the treatment supplemented with Ni and guar gum together with both the soil simulants (around 5% in case of LSS-SP-GG-N and about 4.2% in case of MSS-SP-GG-N), see Figs 6D and 8D. Based on these thermal decomposition results, it can be concluded that guar gum and Ni supplementation together support the formation of vaterite and calcite, thereby augmenting the MICP process.

## 3.5 Flask-condition evaluation of additives

The XRD and microstructural data of consolidated LSS and MSS bricks clearly confirm bacterial induced consolidation in both soil simulants. Peaks representing different phases of calcium carbonate (calcite, aragonite and vaterite) in MSS-SP-N and MSS-SP-GG-N for MSS and in LSS-SP-GG and LSS-SP-GG-N for LSS clearly show the occurrence of bacterial mediated bio-consolidation of martian and lunar bricks. These results serve to demonstrate that supplementation with guar gum and Ni can significantly enhance urease activity in both soil simulants, thereby accelerating the MICP process and leading to considerable increase in compressive strength.

Guar gum has been studied as a binder material by several groups and the following hypotheses have been proposed for its binding action [61, 62]. Firstly, guar gum has electrokinetic charges that ensure formation of strong bonds with soil particles, which is further supplemented by the presence of borate ions. Secondly, the presence of hydroxyl groups in guar gum enables it to form additional hydrogen bonds with soil particles as well as in the solution [61]. Hence it appears likely that similar mechanisms are at work in our experiments as well. However, the precise binding mechanisms with soil and bio-cementation continues to be a topic of active investigation.

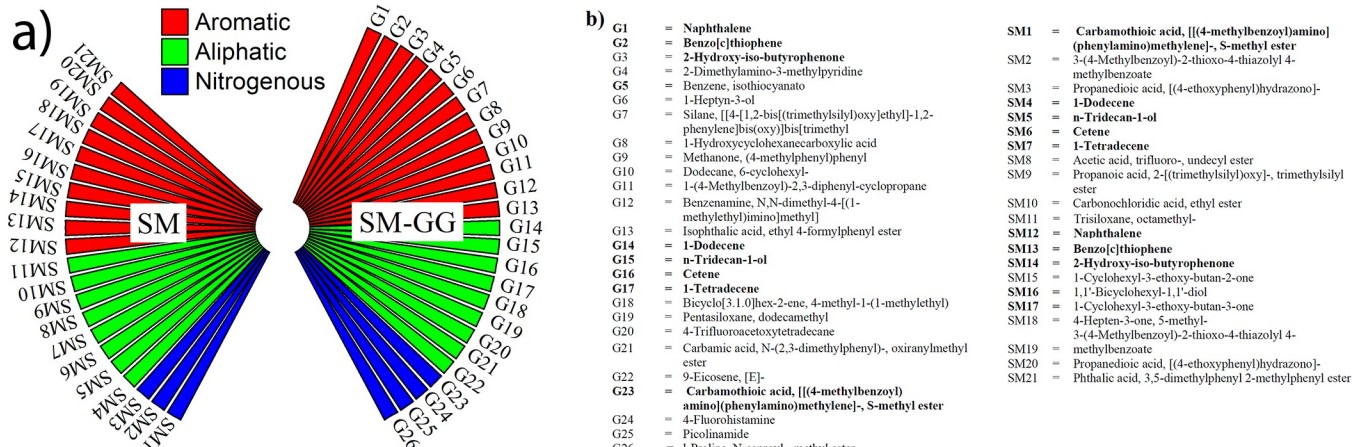

**Fig 9.** a) GCMS profiling for different combination of growth media namely SM without any media additives, SP-GG supplemented with guar gum, inoculated with S. pasteurii under flask condition after 6 days of incubation. SM represent for the treatment of SM without supplementation of guar gum whereas GG for the guar gum supplementation. Hatched column representing common compounds in both treatments. b) Showing name of the metabolites obtained from GCMS.

In addition, bacterial activity relevant to the MICP process due to guar gum supplementation was assessed by GCMS profiling of secondary metabolites. The metabolites in both treatments (SM and SM-GG) were segregated into aliphatic, aromatic and nitrogenous compound, as shown in Fig 9A and the names of the identified metabolites are listed in Fig 9B. In SM-GG treatment, more fused ring aromatic compounds such as methylbenzoyl) amino] (phenylamino)methylene]-, S-methyl ester d, Isophthalic acid, ethyl 4-formylphenyl ester were observed as compared to SM treatment. These additional macromolecules observed in SM-GG treatment could be one of the reasons for increased bacterial metabolic activity leading to enhanced calcium carbonate precipitation and increased strength of space brick. Some important nitrogenous compounds such as 2-Dimethylamino-3-methylpyridine, 4-Fluorohistamine, and Picolinamide were also observed exclusively with SM-GG treatment. There were few common aromatic compounds like Naphthalene, Benzo[b]thiophene, 1-Dodecene, Tridecane and Carbamodithioic acid recorded in both the treatments. Carbamic acid (N-2,3-dimethylphenyl)-,oxiranylmethyl ester) was identified only with guar gum supplementation, which is produced as an intermediate compound during urea hydrolysis. It is reported that organic macromolecules can facilitate the biomineralization process either by providing a structural framework, or regulating its dynamic process such as the nucleation site, orientation and growth of crystal [63–65] etc. as observed with the SP-GG treatment. This constitutes an active area of investigation and has important consequences for the binding mechanics of guar gum.

The biological origins of the increase in compressive strength with NiCl$_2$, and guar gum separately, as well as together, were explored by performing physiological studies on the bacterial strain covering the primary parameters with respect to the ureolytic pathway. Fig 10 shows the change in medium pH, ammonium ion concentration and its effect on bacterial growth for these treatments. Typical lag, log, stationery and death phases of bacterial growth pattern can be seen clearly with and without guar gum and NiCl$_2$ additives (Fig 10A). The lag phase (where bacteria is prepared for multiplication) and the log phase (where the bacteria actually multiply) of bacterial growth were observed to be comparatively longer with SM-GG, and SM-GG-N treatments where guar gum served as the sole carbon source. The lag phase for SM-GG lasted for 6 hours of incubation, as against 2 hours for SM and SM-N treatment. Interestingly, the log

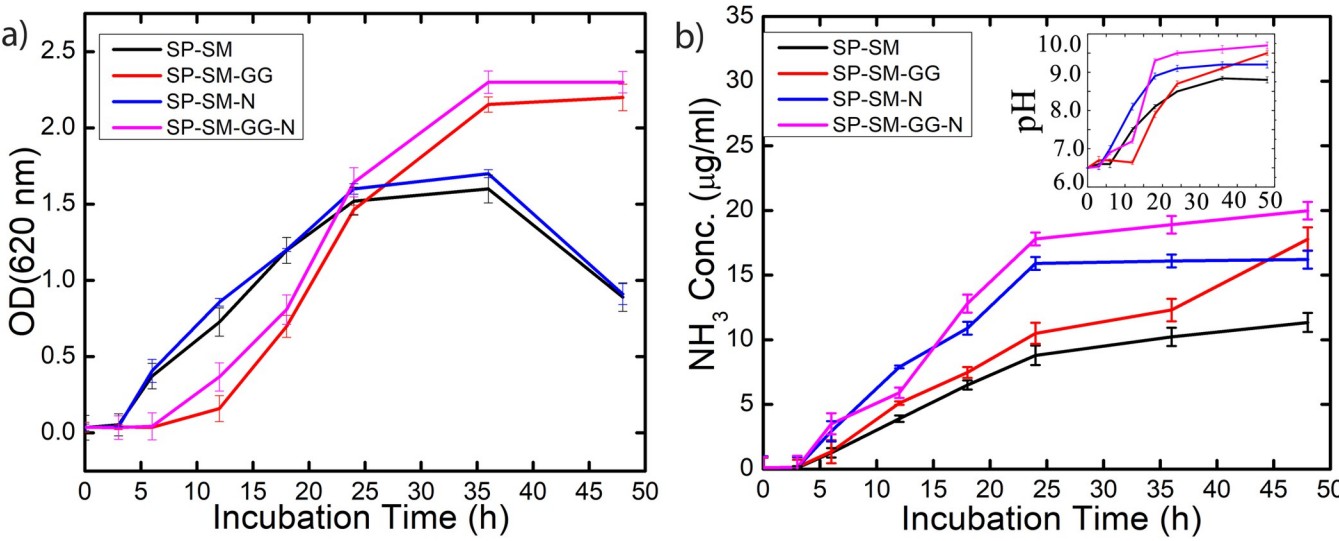

**Fig 10.** Exploration of microbial physiology under flask condition with and without guar gum additive: (a) microbial growth curve (b) temporal evolution of pH (inset) and ammonium ion concentration in growth medium. Legend shows SM; media without any supplementation, SM-GG; SM media supplemented with 1% guar gum, SM-N; SM media with only 10 mM Nickel chloride supplementation, SM-GG-N; SM media supplemented with 1% guar gum and 10 mM Nickel chloride. S. pasteurii was used as inoculum in all cases. (Error bars represents the standard deviation of the data of three independent experiments).

phase was almost 1.5 times longer for treatments with guar gum supplementation in both cases with and without Ni supplementation. The stationary phase (i.e., where cell division and death rate become equal) was smaller and well defined for treatments without guar gum supplementation whereas with guar gum supplementation the stationary phase continued till the end of the experiment duration thus demonstrating sustainability of bacteria.

The supplementation of Ni in the media gave noteworthy results irrespective of the guar gum supplementation. There is a distinct shift seen with Nickel supplementation in the graph for ammonium ion concentration (in SM-N and SM-GG-N treatments as compared to SM and SM-GG treatments). The slope of the plot showed an increase with or without guar gum supplementation depicting an acceleration in the bacterial biochemical process. An increase in ammonium ion concentration was also recorded at the end of the log phase of bacterial growth in these treatments resulting in an incremental shift in the pH (inset image of Fig 10B) of the medium. The supplementation of Nickel increased the ammonium ion concentration values approximately 1.7 times with increase in basicity of the media (Fig 10B). However, the maximum urease activity was observed with the combination of both the additives (SM-GG-N treatment) exploiting their cumulative benefits and thus validating the results obtained for the compressive strength of both the simulants.

## 4. Conclusions

We have shown that bacteria-mediated MICP technique can efficiently consolidate lunar and martian regolith simulant into brick-like structures with promising structural strength. These brick-like structures have been christened 'space-bricks'[44]. Further, we have also presented a casting method, which can be ideal for making various shape structures as well as bricks with consistent strength. A natural polymer (guar gum) as an additive and appropriate concentration of NiCl$_2$ tend to accelerates the overall MICP process thus contributing towards enhancing the strength of the space bricks. Hence this biological approach, coupled with a scalable casting method, towards manufacturing of bricks presents a promising and highly sustainable potential route for *in situ* utilization of structural elements on extra-terrestrial habitats.

## Supporting information

**S1 Fig. Screening of temporal evolution of ammonium ion concentration with the supplementation of different concentration of $NiCl_2$ in flask condition and with soil separately (F denotes flask condition and S in the soil).**
(PDF)

**S1 Data.**
(ZIP)

## Acknowledgments

We thank Dr. Amrit Ambirajan, Research Professor, IISc and former Scientist, ISRO for stimulating discussions and for suggesting the name 'space bricks'. Authors acknowledge GCMS Facility at division of Biological Sciences for GCMS analysis, Centre for Nano Science and Engineering (CeNSE) and Advanced Facility for Microscopy and Microanalysis (AFMM) for SEM and XRD analysis and Solid State and Structural Chemistry Unit (SSCU) of Indian Institute of Science, Bangalore for TGA analysis respectively.

## Author Contributions

**Conceptualization:** Rashmi Dikshit, Aloke Kumar.

**Data curation:** Rashmi Dikshit, Nitin Gupta, Arjun Dey, Koushik Viswanathan, Aloke Kumar.

**Formal analysis:** Rashmi Dikshit, Koushik Viswanathan, Aloke Kumar.

**Investigation:** Rashmi Dikshit.

**Methodology:** Rashmi Dikshit, Nitin Gupta.

**Supervision:** Aloke Kumar.

**Writing – original draft:** Rashmi Dikshit.

**Writing – review & editing:** Nitin Gupta, Arjun Dey, Koushik Viswanathan, Aloke Kumar.

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
