## [Decision Letter · Decision Letter 0]

5 Jan 2022

PONE-D-21-30580

Microbial induced calcite precipitation can consolidate martian and lunar regolith simulants

PLOS ONE

Dear Dr. Kumar,

Thank you for submitting your manuscript to PLOS ONE. After careful consideration, we feel that it has merit but does not fully meet PLOS ONE’s publication criteria as it currently stands. Therefore, we invite you to submit a revised version of the manuscript that addresses the points raised during the review process.

We especially recommend you respond to the somewhat integrated comments of the second reviewer as much as possible: "more in-depth evaluations about the effect of carbonate precipitation on the mechanical properties", " details of the mechanisms triggered by the different preparative routes", and so on.

We look forward to receiving your revised manuscript.

Kind regards,

Il Won Kim

Academic Editor

PLOS ONE

“RD acknowledges funding from Department of Biotechnology, Ministry of Science and Technology, GOI for their grant under BioCare scheme (BT/PR31844/BIC/101/1206/2019). Authors acknowledge thank Indian Space Research Organisation for providing the lunar soil simulant. AK acknowledges partial support from [Grant number- ISTC0415]. We thank Dr. Amrit Ambirajan, Research Professor, IISc and former Scientist, ISRO for stimulating discussions and for suggesting the name ‘space bricks’.”

“RD

BT/PR31844/BIC/101/1206/2019

Department of Biotechnology, Ministry of Science and Technology, GOI

https://dbtindia.gov.in/

AK

ISTC0415

Indian Space Research Organisation

https://www.isro.gov.in/

Reviewers' comments:

Reviewer's Responses to Questions

**Comments to the Author**

1. Is the manuscript technically sound, and do the data support the conclusions?

Reviewer #1: Yes

Reviewer #2: Partly

2. Has the statistical analysis been performed appropriately and rigorously? 

Reviewer #1: I Don't Know

Reviewer #2: N/A

3. Have the authors made all data underlying the findings in their manuscript fully available?

Reviewer #1: Yes

Reviewer #2: Yes

4. Is the manuscript presented in an intelligible fashion and written in standard English?

Reviewer #1: Yes

Reviewer #2: Yes

5. Review Comments to the Author

Reviewer #1: This is an interesting idea and could lead to further developments. It is well written and clearly presented. I do wonder why the authors used a Gram positive bacterium, since a number of calcifying Gram negative bacteria have been investigated and have the ability to produce their own "gums" (extracellular polymeric substances, EPS), which could avoid the need to add yet another ingredient to the growth medium. I should like the authors to include some comparisons with consolidated bricks produced by the other (non microbial) methods that they mention. Are there advantages to this method and, if so, what are they?

Reviewer #2: Dear Authors,

I read with interest your paper about the microbial consolidation of regolith. I must say I expected some more-in-depth evaluations about the effect of carbonate precipitation on the mechanical properties of your material (maybe a grain size vs mechanical quality effect, or something related to the percentage of one calcium carbonate polymorph instead of the others, or the total amount of calcium carbonate precipitated comparing different conditions and so on). In fact, I didn't find what I expected. The paper does not enter into the details of the mechanisms triggered by the different preparative routes but simply compares some samples without trying an explication, a model, a generalization to forecast what will happen by changing one among the variables.

I put some comments directly into the text. Please check typos.

6. PLOS authors have the option to publish the peer review history of their article (what does this mean?). If published, this will include your full peer review and any attached files.

Reviewer #1: No

Reviewer #2: No

---

## [Author Response · Author response to Decision Letter 0]

26 Feb 2022

All the comments raised by the editor and the reviewers are suitably addressed

---

## [Decision Letter · Decision Letter 1]

21 Mar 2022

Microbial induced calcite precipitation can consolidate martian and lunar regolith simulants

PONE-D-21-30580R1

Dear Dr. Kumar,

We’re pleased to inform you that your manuscript has been judged scientifically suitable for publication and will be formally accepted for publication once it meets all outstanding technical requirements.

Kind regards,

Il Won Kim

Academic Editor

PLOS ONE

Additional Editor Comments (optional):

Please check the reviewer’s comment on XRD patterns (vs ‘spectrum’) and amend your text during proofreading.

Reviewers' comments:

Reviewer's Responses to Questions

**Comments to the Author**

1. If the authors have adequately addressed your comments raised in a previous round of review and you feel that this manuscript is now acceptable for publication, you may indicate that here to bypass the “Comments to the Author” section, enter your conflict of interest statement in the “Confidential to Editor” section, and submit your "Accept" recommendation.

Reviewer #2: All comments have been addressed

2. Is the manuscript technically sound, and do the data support the conclusions?

Reviewer #2: Yes

3. Has the statistical analysis been performed appropriately and rigorously? 

Reviewer #2: N/A

4. Have the authors made all data underlying the findings in their manuscript fully available?

Reviewer #2: Yes

5. Is the manuscript presented in an intelligible fashion and written in standard English?

Reviewer #2: Yes

6. Review Comments to the Author

Reviewer #2: Dear Authors,

thank you for your in-depth responses and all the extra material you supplied to support your answers. I must say this is the first time I had the chance of reading such detailed answers with so many additional data that make it clear the paper belongs to a much wider program. For this reason thank you so much.

I recommend to check the paper proofs carefully and change the world "spectrum" that appears in many points of your paper when describing XRPD patterns with "pattern" or "diffraction data" or similar because XRPD is not a spectroscopic technique, but I do not want ask "minor revisions" since you did a lot of work already and I consider the paper deserves to be published on Plos.

7. PLOS authors have the option to publish the peer review history of their article (what does this mean?). If published, this will include your full peer review and any attached files.

Reviewer #2: No

---

## [Editor Report · Acceptance letter]

29 Mar 2022

PONE-D-21-30580R1 

Microbial induced calcite precipitation can consolidate martian and lunar regolith simulants 

Dear Dr. Kumar:

I'm pleased to inform you that your manuscript has been deemed suitable for publication in PLOS ONE. Congratulations! Your manuscript is now with our production department. 

Kind regards, 

on behalf of

Professor Il Won Kim 

Academic Editor

PLOS ONE